# Hybrid LLM: Cost-Efficient and Quality-Aware Query Routing

**Dujian Ding**[1]*, **Ankur Mallick**[2], **Chi Wang**[2], **Robert Sim**[2], **Subhabrata Mukherjee**[3]†,
**Victor Ruhle**[2], **Laks V. S. Lakshmanan**[1], **Ahmed Awadallah**[2]
[1] University of British Columbia      [2] Microsoft      [3] Hippocratic AI
dujian.ding@gmail.com    ankurmallick@microsoft.com

## ABSTRACT

Large language models (LLMs) excel in most NLP tasks but also require expensive cloud servers for deployment due to their size, while smaller models that can be deployed on lower cost (e.g., edge) devices, tend to lag behind in terms of response quality. Therefore in this work we propose a hybrid inference approach which combines their respective strengths to save cost and maintain quality. Our approach uses a router that assigns queries to the small or large model based on the predicted query difficulty and the desired quality level. The desired quality level can be tuned dynamically at test time to seamlessly trade quality for cost as per the scenario requirements. In experiments our approach allows us to make up to 40% fewer calls to the large model, with no drop in response quality.

## 1 INTRODUCTION

Large language models (LLMs) have become the dominant force in natural language processing in recent years (Zhao et al., 2023). Their impact has been especially striking in generative applications where it has extended beyond standard language understanding and question-answering benchmarks like (Hendrycks et al., 2020; Srivastava et al., 2022) to several successful real-world deployments. These include the wildly popular ChatGPT (OpenAI, b) and several other chatbots (Zheng et al., 2023) powered by different LLMs (Taori et al., 2023; Touvron et al., 2023; OpenAI, 2023), which allow users to engage in natural language conversations and obtain informative responses on a range of practically useful tasks like creative writing, translation, code completion, etc. An important added attraction of these models is their accessibility. Users can input queries and receive responses in natural language, without any specialized data or code, and this is what has created such a widespread demand for their services across regions, professions, and disciplines.

The best performing LLMs are based on the transformer architecture of (Vaswani et al., 2017) and generally have tens of billions of parameters. E.g., Alpaca (Taori et al., 2023) has 13 billion parameters, the best version of Llama-2 (Touvron et al., 2023) has 70 billion parameters, while OpenAI's GPT-3.5 (OpenAI, a) and GPT-4 (OpenAI, 2023) are rumored to be much larger. Their enormous size and the autoregressive nature of text generation in their transformer architectures means that these models typically have a high compute and memory requirement that can only be met by expensive cloud servers (Yu et al., 2022). This can potentially impose an enormous cost on developers and users as more LLM-based services are introduced. In response to this there has been a surge of interest in designing smaller, cost-effective LLMs – e.g., (Touvron et al., 2023) provides multiple versions of Llama-2, with the smallest having only 7 billion parameters, small enough to run on a laptop[1], while the smallest offering of Google's Palm-2 model can even run on mobile devices[2]. However empirical evaluations in (Chung et al., 2022; Touvron et al., 2023) as well as our own evaluation in Figure 1a show that smaller models generally lag behind in terms of response quality.

---

*work performed during internship at Microsoft

†work performed while at Microsoft

[1]https://github.com/microsoft/Llama-2-Onnx

[2]https://blog.google/technology/ai/google-palm-2-ai-large-language-model/

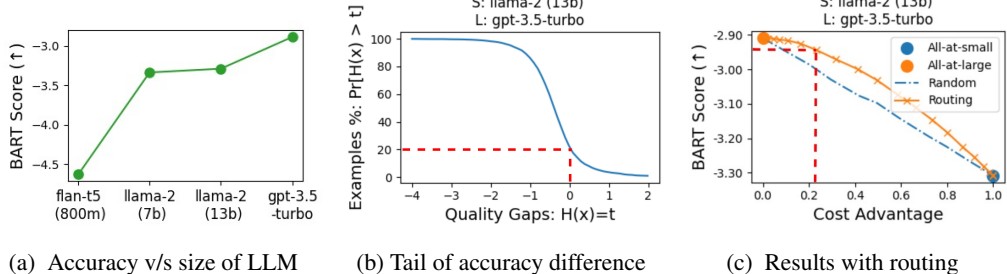

(a) Accuracy v/s size of LLM     (b) Tail of accuracy difference     (c) Results with routing

Figure 1: We use a dataset of natural language queries from a range of tasks like question answering, summarization, information extraction, etc. (See Section 4 for details). We observe that (a) smaller models generally give poorer response quality or lower BART score (Yuan et al., 2021), (b) Llama-2 (13b) outperforms GPT-3.5-turbo on around $20\%$ examples, and (c) our router can make $22\%$ fewer calls to GPT-3.5-turbo (cost advantage) with $1\%$ drop in response quality (BART score).

Faced with this tradeoff between response quality and inference cost, we propose a hybrid inference approach which provides the best of both worlds. Our approach is motivated by the observation that most tasks for which LLMs are useful, like creative writing, translation, code completion, etc., include a range of queries of different difficulty levels and there is always a subset of "easy" queries for which responses of a small (inexpensive and weak) model may be comparable to, and sometimes even better than those of a large (expensive and powerful) model. This is also illustrated in Figure 1b where we plot the tail of the quality gap (defined in Section 3) between the 13 billion parameter version of Llama-2 and OpenAI's GPT-3.5-turbo, the model that powers ChatGPT. Quality gap is non-negative for examples where the response quality of Llama-2 is comparable to or better than that of GPT-3.5-turbo which is the case for around $20\%$ queries in our dataset (described in Section 4).

We leverage this insight to train a router that takes a large model and a small model as input, and learns to identify these easy queries as a function of the desired level of response quality, while taking into account the generative nature of tasks, inherent randomness in LLM responses, and response quality disparity between the two models. At test time, the router seamlessly adjusts to different response quality requirements and assigns the corresponding "easy" queries to the small model, leading to significant inference cost reduction with minimal drop in response quality. In Figure 1c our router assigns $22\%$ of queries to Llama-2 (13b) [3] with less than $1\%$ drop in response quality measured in BART scores (Yuan et al., 2021). The gains are even higher for pairs where the small model is closer in terms of response quality to the large model (see Section 4).

With the explosion in the complexity and costs of LLM deployments, small companies and individual consumers, have started to rely on the pre-existing LLMs hosted on platforms like HuggingFace (HuggingFace) and OpenAI (OpenAI, c). In this context, our hybrid inference approach can reduce the costs incurred by both consumers and platform owners because a) consumers can use it to route easy queries to small models hosted on their edge devices (laptops/smartphones) and only call the API for the more complex queries (illustrated in Figure 2) and b) platform owners can automatically route queries to lower cost models at the backend without affecting the user experience, as long as the response quality levels are maintained. Thus our hybrid inference approach offers a flexible and cost-effective solution for harnessing the full potential of LLMs while accommodating diverse cost budgets and quality requirements.

The main technical contributions of this work are: a) we are the first to explore cost-effective and quality-aware hybrid LLM inference, b) we design a novel query router which routes queries based on an estimate of the response quality gap between models (Section 3.1), c) we incorporate uncertainty due to randomness in LLM responses in our router design to improve performance (Section 3.2), d) we identify challenges for our router when the small model is significantly weaker than the large model and introduce a novel data transformation to address this issue (Section 3.3), and e) we provide extensive experimental results (Section 4) on a large benchmark dataset of real world natural language queries and responses (Jiang et al., 2023) thereby demonstrating the value of the

---

[3] We term the fraction of queries routed to the small model as the *cost advantage* (see §2.3)

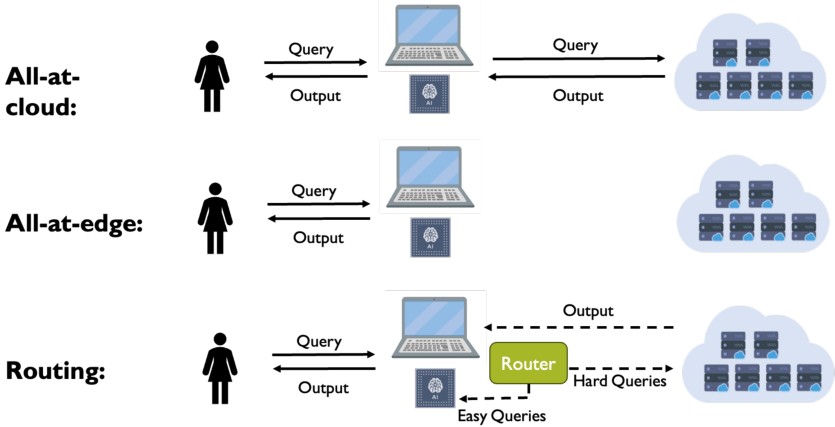

Figure 2: Routing between edge and cloud.

approach and its superiority over baseline approaches, enabling LLM providers and consumers to cost-efficiently enable LLM-backed experiences.

## 2 PROBLEM FORMULATION

### 2.1 RELATED WORK

**Large Language Models (LLMs).** The advent of LLMs has led to a paradigm shift in the study of natural language processing (NLP), computer vision, information retrieval, and other domains(Menghani, 2023; Chen et al., 2023; Jiang et al., 2023). The impressive effectiveness and generalizability of LLMs has come at the price of a drastic increase in LLM sizes (Treviso et al., 2023) and consequent challenges, including huge amounts of computational resources and data required to train, and prohibitive expenses at both training and deployment stages (Bender et al., 2021).

**Efficient Machine Learning (ML) Inference.** LLMs belong to a class of models called *foundation models* (Bommasani et al., 2021) – models that are trained once and can then be used to serve a wide variety of tasks. As such, we expect inference cost to dominate the overall cost of such models and hence focus on works that reduce the cost of ML inference (Menghani, 2023). Common techniques for efficient ML inference include model pruning (LeCun et al., 1989), quantization (Jacob et al., 2018), knowledge distillation (Hinton et al., 2015), and Neural Architecture Search (Elsken et al., 2019). Such *static* efficiency optimizations typically produce a *fixed* model with lower inference cost and lower accuracy compared to the large model which may not suffice for foundation models like LLMs, whose core premise is that the same model will serve a range of tasks, each with its own accuracy/cost constraints. This is already manifesting in inference platforms described in Section 1 which need more dynamic optimizations to meet the demands of all users.

**Hybrid ML Inference.** Recent works (Kag et al., 2022; Ding et al., 2022) have introduced a new inference paradigm called hybrid inference which uses two models of different sizes instead of a single model for inference. The smaller model (e.g. Llama-2 (Touvron et al., 2023)) generally has lower inference cost but also lower accuracy than the larger model (e.g. GPT-4 (OpenAI, 2023)). The key idea is to identify and route easy queries to the small model so that inference cost can be reduced while maintaining response quality. By tuning a threshold on query difficulty we can *dynamically* trade off quality and cost for the same inference setup. (Kag et al., 2022) study this setup for image classification and propose to train the small model, large model, and router from scratch. However LLM training is expensive and retraining LLMs from scratch for every scenario goes against the very premise of inference with pre-trained foundation models. Moreover text generation (Iqbal & Qureshi, 2022) is often more ambiguous and challenging than image classification due to which novel techniques are required for effective hybrid LLM inference for text generation.

**Inference with Multiple LLMs.** Some recent works use multiple LLMs for inference but these approaches typically call more than one LLM for a single query that can incur significant compu-

tational overheads. Specifically (Jiang et al., 2023) calls an ensemble of LLMs at inference time due to which the inference cost will be proportional to the number of models in the system. (Chen et al., 2023) performs inference by sequentially calling LLMs until one has a high confidence score exceeding the predefined threshold. Our work provides high quality responses while always calling a single LLM for all queries and will thus incur much lower costs than both of these works on average. Speculative decoding (Leviathan et al., 2023; Kim et al., 2023) speeds up decoding of expensive LLMs by invoking small decoders on the "easy" decoding steps. Instead, in our work we are interested in query routing which assigns "easy" queries to small models to reduce overall inference costs while maintaining high performance. While these two approaches have different goals, an interesting line of future work would be to combine these to achieve further cost reduction.

## 2.2 Problem Setting

We extend the hybrid ML paradigm to LLM inference by routing queries between two models with different inference costs and accuracy. We use $\mathcal{X}$ and $\mathcal{Z}$ to denote the input query space and the set of all possible output responses respectively. Let $L : \mathcal{X} \to \mathcal{Z}$ denote the large model and $S : \mathcal{X} \to \mathcal{Z}$ denote the small model. Formally, the objective in our paradigm is to learn a router $r : \mathcal{X} \to \{0, 1\}$ such that each user query $x \in \mathcal{X}$ is routed to the small model $S(x)$ if $r(x) = 0$, and to the large model $L(x)$, otherwise. Note that we always route each query to a single LLM at inference time as opposed to using an ensemble (Jiang et al., 2023) or a cascade (Chen et al., 2023) of LLMs, which may call multiple LLMs to resolve a single query and incur significant computational overheads.

## 2.3 Evaluation Metric

**Response Quality** Automatic evaluation for text generation is a challenging and widely studied problem. Traditional metrics, such as BLEU and ROUGE, initially designed for machine translation and summarization, have been found to be of limited concordance with human judgment and restricted applicability across diverse NLP tasks (Blagec et al., 2022). Significant research efforts have been devoted to implementing task-agnostic evaluation metrics with neural networks. GPT-ranking (Jiang et al., 2023), as a representative example, employs GPT models (e.g., GPT-4 (OpenAI, 2023)) to provide relative rankings between pairs of generated outputs. In spite of the high correlation with human perception, GPT-ranking suffers from high computational costs and inability to distinguish between examples with the same ranking. Instead, we use the BART score (Yuan et al., 2021) to evaluate response quality of different models since (1) it is inexpensive to compute in comparison to LLM-based metrics such as GPT-ranking, and (2) it has been shown in prior work (Jiang et al., 2023) that this metric correlates well with the ground truth. We also provide a case study in Appendix C.2 to empirically justify using BART score as the response quality metric. We use $q(z)$, $q : \mathcal{Z} \to \mathbb{R}$ to denote the BART score (response quality) of model responses $z \in \mathcal{Z}$.

**Cost Advantage** The absolute costs of running a model may not be known *a priori*, and may be expressed using a variety of metrics, including latency, FLOPs, energy consumption, etc. In LLM inference, however, each of these metrics is affected by several underlying confounders such as different prompt templates, hardware capability, network connectivity, etc. Moreover different platforms/users may be interested in different metrics. However the common underlying assumption in this and previous works on efficient ML inference is that smaller models are more efficient than larger models and therefore we expect to obtain an improvement in all the metrics by routing more queries to the smaller model. Hence we define *cost advantage* as the percentage of queries routed to the smaller model. Note that the notion *cost advantage* has been used as a generic efficiency metric in previous hybrid ML work (Kag et al., 2022), where it is termed as *coverage*.

## 3 Hybrid LLM Inference

**Easy Queries.** We refer to queries for which the response quality of the small model is close to the response quality of the large model as "easy" queries. The goal of our hybrid inference framework is to identify the easy queries and route them to the small model thereby ensuring significant inference cost reduction without much drop in response quality. Note that the easy queries as defined here, need not necessarily be queries that are easy/inexpensive to respond to, they are just queries for

which the small model can match up to the large model. Examples of easy and hard queries as per this definition are provided in Appendix C.1.

**Quality Gap.** We define quality gap of a query $x$ as $H(x) := q(S(x)) - q(L(x))$ i.e. the difference in quality of the small model's response $S(x)$ and the large model's response $L(x)$. The quality gap is a random variable since LLM responses are typically non-deterministic. This is illustrated in Figure 3 below where the blue and orange plots correspond to the distribution of responses from FLAN-t5 (800m) [4] (Chung et al., 2022) and Llama-2 (13b) (Touvron et al., 2023) for a single query.

**Proposed Orchestration Framework.** Queries are routed using a BERT-style encoder model (e.g., DeBERTa (He et al., 2020)) which is trained on a dataset of representative queries and learns to predict a score. Since the router is an encoder model, a single pass of the query through it is sufficient to generate the score and we assume that the cost of this step is negligible compared to the cost of running autoregressive decoding using the large model $L(x)$ (Sun et al., 2019). Thus, we expect that using the router to route queries to the small model will not detract significantly from the realizable cost advantage.

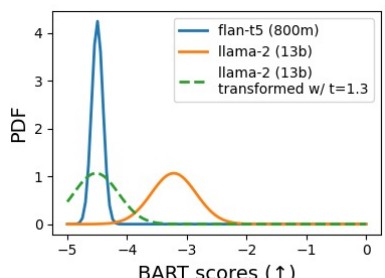

Figure 3: Response quality distribution for FLAN-t5 (800m) and Llama-2 (13b) on the query **"How to identify the index of median?"** measured in BART scores. Llama-2 (13b) with transformation significantly overlaps with FLAN-t5 (800m).

**Router Score.** We design the router score to be large for easy queries as defined above. Intuitively, an estimate of $\Pr[H(x) \geq 0]$ is a suitable candidate since a large value of $\Pr[H(x) \geq 0] = \Pr[q(S(x)) \geq q(L(x))]$ corresponds to queries for which there is a high likelihood that the response quality of the small model will be at least as high as that of the large model. However we show below that in scenarios where the large model is significantly more powerful than the small model i.e. $q(S(x)) << q(L(x))$ in general, one can train more effective routers by relaxing the definition of easy queries to $\Pr[H(x) \geq -t] = \Pr[q(S(x)) \geq q(L(x)) - t]$ for an appropriate $t > 0$. At test time we achieve the desired performance accuracy tradeoff by tuning a threshold on the score and routing queries with score above the threshold to the small model. For a router with parameters $w$, we denote router score by $p_w(x)$, $p_w : \mathcal{X} \to [0, 1]$. We discuss different router score designs in the rest of this section assuming a training set of $N$ queries $x_1, \ldots, x_N$.

## 3.1 DETERMINISTIC ROUTER

Previous work on hybrid ML (Ding et al., 2022; Kag et al., 2022) assumes that neural models are deterministic functions mapping input features to a single point in the output space. To realize this for LLMs, we sample a *single* response per query from each model. We assign boolean labels $y_i^{\text{det}} = \mathbb{1}[q(S(x_i)) \geq q(L(x_i))]$ to each training query $x_i$ with the BART score as the quality function $q(.)$. The router is trained by minimizing the binary cross-entropy loss (Ruby & Yendapalli, 2020).

$$\mathcal{L}(w) = -\frac{1}{N} \sum_{i=1}^{N} \left( y_i^{\text{det}} \log(p_w(x_i)) + (1 - y_i^{\text{det}}) \log(1 - p_w(x_i)) \right) \tag{1}$$

Observe that the assigned labels $y_i^{\text{det}}$ can be viewed as an estimate for $\Pr[H(x_i) \geq 0]$ given a single response per query from each model and thus minimizing the above loss encourages the router score $p_w(x)$ to be close to $\Pr[H(x) \geq 0]$ for test queries. We refer to this deterministic router as $r_{\text{det}}$.

## 3.2 PROBABILISTIC ROUTER

The determinism assumption can be justified for tasks where the ground truth labels are often explicit and unique such as image classification (Masana et al., 2022) and video segmentation (Yao et al., 2020). When it comes to NLP tasks, however, there is usually no single best answer due to the intrinsic ambiguity and complexity of natural languages. LLMs are widely used as non-deterministic generators to capture the intrinsic uncertainty of NLP tasks, as shown in Figure 3 (ignore the dashed

---

[4]We use the FLAN-t5-large model from https://huggingface.co/google/flan-t5-large.

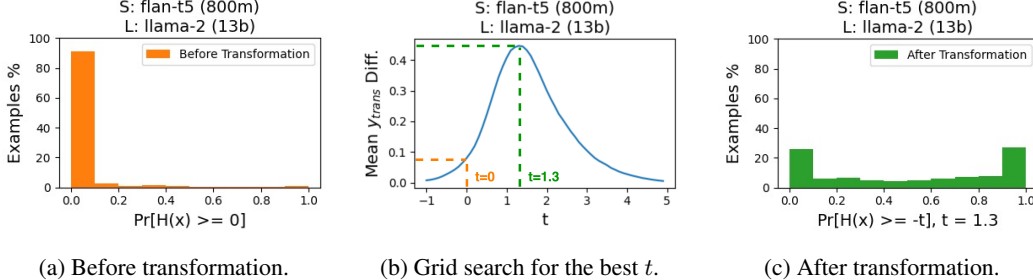

Figure 4: Effect of data transformation on labels for training the router.

curve for now). The non-determinism mainly comes from the randomness in the decoding phase. Users typically control the level of uncertainty by choosing different decoding strategies such as nucleus sampling (Holtzman et al., 2019), as well as the values of the hyper-parameter *temperature*. Intuitively, higher temperature values result in a higher level of randomness and diversity among the generated responses. For black-box LLM APIs such as GPT-4 (OpenAI, 2023), it has been observed that even upon setting temperature to the minimum value 0, it can still provide different responses for the same input queries. The underlying mechanism is still an open problem while a recent study hints at the instability of the MoE backbone (Skyward, 2023).

We propose to incorporate the uncertainty into the router training loss by relaxing the hard labels $y_i^{\text{det}} \in \{0, 1\}$ to the soft labels $y_i^{\text{prob}} := \Pr[H(x_i) \geq 0] = \Pr[q(S(x_i)) \geq q(L(x_i))] = \mathbb{E}[\mathbb{1}[q(S(x_i)) \geq q(L(x_i))]]$ where $\mathbb{E}$ denotes the expectation. In practice, we estimate expectation by sampling 10 responses from each model and computing the sample average of the corresponding indicator function values. Observe that the hard label $y_i^{\text{det}}$ is a higher-variance estimate of $\mathbb{E}[\mathbb{1}[q(S(x_i)) \geq q(L(x_i))]]$ (since it is obtained from a single sample) and hence we expect improved performance of the probabilistic router with the following training loss, referred to as $r_{prob}$.

$$\mathcal{L}(w) = -\frac{1}{N} \sum_{i=1}^{N} \left( y_i^{\text{prob}} \log(p_w(x_i)) + (1 - y_i^{\text{prob}}) \log(1 - p_w(x_i)) \right) \quad (2)$$

### 3.3 PROBABILISTIC ROUTER WITH DATA TRANSFORMATION

While so far we have designed scores that try to estimate $\Pr[H(x) \geq 0]$, we observe that the empirical estimate of $\Pr[H(x_i) \geq 0] = \mathbb{E}[\mathbb{1}[q(S(x_i)) \geq q(L(x_i))]]$ tends to be extremely small when the large model is significantly more powerful than the small model (0 for almost 90% of the queries in Figure 4a with Flan-t5 (800m) as the small model and Llama-2 (13b) as the large model). Because $q(S(x)) << q(L(x))$ for most queries in this case, it provides an extremely weak signal for training using Equation (2) and as shown in Section 4 both $r_{det}$ and $r_{prob}$ fail to provide much improvement over random query assignment in this case.

Traditional approaches for learning with imbalanced data have their own shortcomings (Krawczyk, 2016). Moreover our goal is to only design a router that can reduce inference cost while maintaining response quality as much as possible and so we are not tied to a particular definition of class labels to achieve this. We leverage this flexibility to introduce new labels $y_i^{\text{trans}}(t) := \Pr[H(x_i) \geq -t] = \Pr[q(S(x_i)) > q(L(x_i)) - t]$ for some $t > 0$. Since $-t < 0$, $\Pr[H(x) \geq -t] \geq \Pr[H(x) \geq 0]$ by definition of the tail distribution and so we expect this relaxation to provide a stronger signal for router training while still allowing us to identify the easy queries i.e. those queries for which $q(S(x))$ has a high likelihood of being close to $q(L(x))$ ($q(S(x)) > q(L(x)) - t$). Visually, this corresponds to comparing the distribution of the small model's response with a *shifted* distribution of the large model's response to a query (dotted curve in Figure 3).

Now the question is *how to choose the best relaxation $t$?* Given that tail probability $\Pr[H(x) \geq -t]$ lies in $[0, 1]$, we choose $t$ by maximizing the average pairwise differences between the transformed labels to push them as far apart as possible and provide a strong signal for training. Thus we set,

$$t^* = \arg\max_t \frac{1}{N^2} \sum_{(i, i')} \mid y_i^{\text{trans}}(t) - y_{i'}^{\text{trans}}(t) \mid \quad (3)$$

Table 1: Cost advantage v.s. performance drop for model pairs of different performance gaps. Performance drops are computed w.r.t. the *all-at-large* baseline.

| Cost Advantage (%) | Response Quality (BART Score) Drop w.r.t all-at-large (%) | | | | | | | | |
|---|---|---|---|---|---|---|---|---|---|
| | S: Llama-2 (7b) | | | S: Llama-2 (13b) | | | S: FLAN-t5 (800m) | | |
| | L: Llama-2 (13b) | | | L: GPT-3.5-turbo | | | L: Llama-2 (13b) | | |
| | $r_{det}$ | $r_{prob}$ | $r_{trans}$ | $r_{det}$ | $r_{prob}$ | $r_{trans}$ | $r_{det}$ | $r_{prob}$ | $r_{trans}$ |
| 10 | 0.1 | **-0.1** | 0.1 | 0.1 | **-0.1** | 0.2 | 2.3 | 2.2 | **2.1** |
| 20 | 0.1 | **0.0** | **0.0** | 1.0 | **0.8** | 0.8 | 5.8 | 5.8 | **4.7** |
| 40 | 0.2 | 0.1 | **0.0** | 3.5 | 3.4 | **2.9** | 13.8 | 13.1 | **10.3** |

We currently solve the above optimization problem via grid-search and leave more sophisticated approaches for future work. We plot the optimization objective for different values of $t$ for our training dataset in Section 3.3 and show the distribution of transformed labels $y_i^{\text{trans}}(t^*)$ in Figure 4c. As we see, the distribution is significantly more balanced now and we expect the resulting router to be much more effective. We train the router by minimizing the loss, referred to as $r_{trans}$.

$$\mathcal{L}(w) = -\frac{1}{N} \sum_{i=1}^{N} \left( y_i^{\text{trans}}(t^*) \log(p_w(x_i)) + (1 - y_i^{\text{trans}}(t^*)) \log(1 - p_w(x_i)) \right) \tag{4}$$

# 4 EVALUATION

## 4.1 EVALUATION SETUP

**Dataset.** We use the MixInstruct dataset from (Jiang et al., 2023) to evaluate the effectiveness of different routing strategies. MixInstruct consists of a wide range of tasks (e.g., question answering, summarization, information extraction) and enables us to train a generic router that will be effective across different scenarios. We present additional information about this dataset in Appendix B.

**Router Model.** We use DeBERTa-v3-large (He et al., 2020) (300M) as the backbone to train our routers. We train each router with the corresponding loss from Section 3 for 5 epochs and use the validation set to choose the best checkpoints for final evaluation. All experiments are conducted with 1 NVIDIA A100 GPU of 80GB GPU RAM. We have made our source code available at https://github.com/m365-core/hybrid_llm_routing.

**Evaluation Measures.** We use BART score (Yuan et al., 2021) as the quality metric and use fraction of queries routed to the small model (*cost advantage*) as the efficiency metric (see Section 2.3).

**Baselines.** We consider three baselines: *all-at-large*, *all-at-small*, and *random*. *All-at-large* routes all queries to the large model, while *all-at-small* routes all queries to the small model. *Random* generates a random number in [0,1] and selects the large model if it is below the probability threshold.

**Experiments.** We investigate all three routers: $r_{det}$, $r_{prob}$, and $r_{trans}$. We select candidate model pairs from FLAN-T5 (800m), FLAN-T5 (11b), Llama-2 (7b), Llama-2 (13b), and GPT-3.5-turbo for our experiments. At test time the trained router ($r_{det}$, $r_{prob}$, or $r_{trans}$) takes a threshold value as input and routes all queries with router score higher than the threshold to the small model as these are the easy queries. We evaluate the router performance in Section 4.2 in terms of both BART score and *cost advantage* (Figure 5 and Table 1), validate that the router is indeed routing easy queries to the small model in Section 4.3, demonstrate that our routers are of negligible compute overhead in Appendix A.1, show how to choose routing thresholds in practice in Appendix A.2, evaluate the effectiveness of our routers using a response quality metric other than the BART score in Appendix A.3, and test the generalizability of routers across model pairs in Appendix A.4.

## 4.2 ROUTER PERFORMANCE RESULTS

**Small performance gap.** LLMs of the same architectures are observed to be of small performance gap such as Llama-2 (7b) v.s. Llama-2 (13b), as seen in Figure 5a. In this case, by trading little to

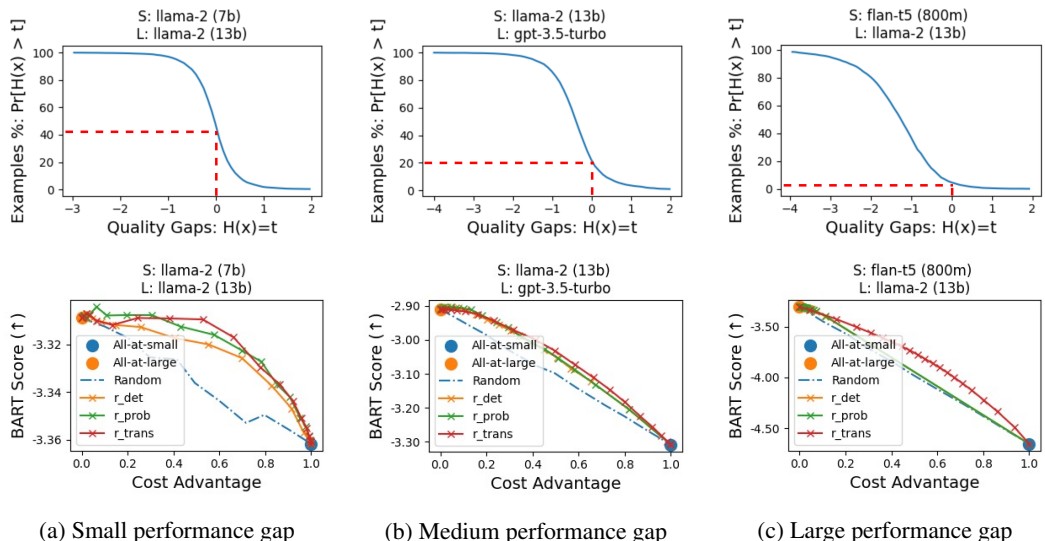

Figure 5: Error-cost tradeoffs achieved by $r_{det}$, $r_{prob}$, and $r_{trans}$ for different performance gaps.

no performance drop, we show that (1) the deterministic router $r_{det}$ can achieve good cost advantages, (2) $r_{prob}$ consistently improves $r_{det}$, and (3) $r_{trans}$ is able to match or slightly improve the performance of $r_{prob}$. Numerical comparison results are summarized in Table 1. $r_{det}$ routes 20% (40%) queries to the small model i.e. Llama-2 (7b) with only 0.1% (0.2%) drop in response quality w.r.t. the *all-at-large* baseline. Impressively $r_{prob}$ and $r_{trans}$ achieve 20% cost advantages without any quality drop, and $r_{trans}$ is able to achieve even 20% cost advantage without quality drop, which can be attributed to these methods capturing the non-deterministic nature of LLMs.

**Medium performance gap.** Often there is only a moderate performance gap between leading open-source LLMs like Llama-2 (13b) and state-of-the-art commodified LLMs, such as GPT-3.5-turbo (Figure 5b). In this case, all our routers deliver reasonable cost advantages with acceptable quality drop. The effectiveness order between $r_{det}$, $r_{prob}$, and $r_{trans}$ resembles that in the small quality gap case. All routers achieve 20% (40%) cost advantage with $\leq 1\%$ ($\leq 4\%$) quality drop (Table 1). In the 40% cost advantage regime, $r_{prob}$ slightly outperforms $r_{det}$ and $r_{trans}$ improves $r_{prob}$ by 0.5% in terms of quality drop.

**Large performance gap.** In the edge-cloud routing scenarios, edge devices often have very limited resources and can only support small models of limited quality, which can be significantly outperformed by large models deployed on the cloud. We investigate how to effectively route queries with LLM pairs of large performance gaps, such as FLAN-t5 (800m) and Llama-2 (13b) (Figure 5c). Non-trivial routing is challenging in this situation since the large model dominates for a majority of examples. Both $r_{det}$ and $r_{prob}$ perform marginally better than the random routing baseline. In contrast, $r_{trans}$ can still effectively distinguish relatively easy queries from the harder ones. $r_{trans}$ achieves 40% cost advantages with 10.3% quality drop, which is 3.5% and 2.8% lower than $r_{det}$ and $r_{prob}$ respectively (Table 1).

In the course of these experiments we made two interesting observations. Firstly, when the cost advantage is modest (e.g., 10%) and the LLM performance gaps are not large (e.g., Llama-2 (7b) v.s. Llama-2 (13b)) $r_{prob}$ is able to achieve even better performance than *all-at-large* which leads to the "negative quality drops" in Table 1. This is because, as seen from the large value of $\Pr[H(x) \geq 0]$ in the tail distribution in Figure 5, the response quality of the small model may be higher than that of the large model for several queries and by routing these queries to the small model, the router is able to even beat *all-at-large*. Secondly, for lower cost advantages ($\leq 10\%$) and small or moderate LLM performance gaps $r_{trans}$ can be slightly outperformed by $r_{det}$ or $r_{prob}$. This might be due noise in the estimation of the relaxation parameter $t$ from sample averages instead of expectation in Equation (3) and from the grid search process leading to suboptimal settings of $r_{trans}$. However we clearly see that in more challenging routing scenarios with high cost advantage targets or large LLM

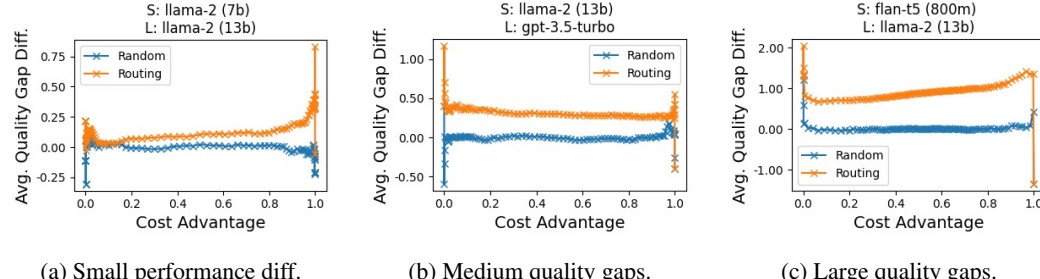

Figure 6: Difference between average quality gap of queries routed to the small and large models with different performance gaps.

performance gaps, both $r_{det}$ and $r_{prob}$ have difficulty in correctly routing queries, and $r_{trans}$ starts to dominate due to the benefits of the data transformation.

### 4.3 ROUTER VALIDATION RESULTS

We also validate that the router is functioning as intended, that is, routing easy queries to the small model and hard queries to the large model. To see this, in Figure 6 we plot the difference between the average quality gaps of queries routed to the small model and those routed to the large model for our router and the random baseline v/s different values of cost advantages (i.e., the fraction of queries routed to the small model). Since the random baseline randomly assigns queries the average difference is nearly always zero. However our router routes easy queries i.e. queries with large quality gap $(q(S(x)) - q(L(x)))$ to the small model and queries with small quality gap to the large model. Hence the difference between the average quality gaps always has a significant positive value indicating that more easy queries are routed to the small model than to the large model in our approach as compared to the random assignment approach at all cost advantages.

## 5 DISCUSSION AND CONCLUSION

Motivated by the need to optimize the trade-off between LLM inference costs and response quality, we have presented a hybrid inference approach based on quality-aware query routing. We train a router to discriminate between "hard" and "easy" queries, enabling the LLM provider to make cost-efficient decisions about which model should serve a given query. Our experimental results on a variety of state-of-the-art LLMs of varying sizes show that such an optimization is possible and that we can realize cost advantages of up to 40% with no significant drop in response quality.

To the best of our knowledge, this is the first work exploring the possibilities of cost-effective and quality-aware query routing between LLMs. We identify several important extensions for future work: (1) **Task-aware routing**. Our current routers make routing decisions purely based on query inputs. To improve routing effectiveness, we can provide more informative signals which help routers distinguish easy queries from the hard ones, such as task labels for query examples and can also identify tasks which may be more suited to routing for a given pair of LLMs. (2) **Generalizing to $N$-model routing**. Modern MLaaS platforms typically host a large number of LLM instances of the same or different configurations to efficiently serve users in different scenarios. This naturally forms a more challenging routing problem with richer optimization opportunities (e.g., load balancing) (3) **Out-of-distribution (OOD) generalization**. In this work, the model pair and data distribution is fixed across training and testing. In the real-world it may be cumbersome/infeasible to train a new router for every new model pair and for every new data distribution. Therefore there is a need for techniques to generalize our approach to changes in the model pair and/or data distribution at test time. (4) **Novel evaluation metrics**. Effective evaluation metrics are critical to train high-quality routers. It is intriguing to see how to develop metrics of higher human-judgment correlation and to which extent it will improve the routing performance.

ACKNOWLEDGMENTS

The authors would like to thank Daniel Madrigal Diaz, Mirian del Carmen Hipolito Garcia, Chen Dun, Guoqing Zheng, Menglin Xia, Wen Xiao, and Jieyu Zhang for helpful discussions.

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

## A    ADDITIONAL EXPERIMENTS

### A.1    ROUTER LATENCY

We measure the latency of our router and compare it to the latency of the different LLMs – Flan-t5 (800m), Llama-2 (7b), and Llama-2 (13b) that we use in our experiments for generating responses to user queries. Note that the latency of all the routers $r_{det}$, $r_{prob}$, and $r_{trans}$ will be the same since they use the same model (DeBERTa-v3-large (He et al., 2020)) and are just trained differently. Also, we do not measure the latency of GPT-3.5-turbo since its responses are generated by querying the OpenAI API (OpenAI, c) as the model weights are not publicly available due to which it is not possible to disentangle the inference latency from the network latency, queueing delay, latency of the API call, etc. However we note that the inference latency of all other LLMs we consider is significantly larger than that of the router (see Table 2) and therefore we expect the same to hold for GPT-3.5-turbo as well.

The latency results are reported in Table 2 where we measure the average latency per query averaged over 200 randomly chosen queries from our dataset (confidence bounds correspond to one standard error). As expected the router processes queries significantly faster than all the LLMs (nearly $10\times$ faster than the fastest LLM – FLAN-t5(800m)). This is both due to its smaller size (300m parameters) and the fact that it performs a single forward pass over the query to generate the score while the LLMs generate the response token-by-token in an autoregressive fashion due to which the inference latency is proportional to the response length. Thus the router adds minimal overhead to the inference cost due to its small size and extremely low latency.

Table 2: Latency Values for Different Models.

| Model | Latency (seconds) |
|---|---|
| Router | $0.036 \pm 0.002$ |
| FLAN-t5 (800m) | $0.46 \pm 0.039$ |
| Llama-2 (7b) | $7.99 \pm 0.15$ |
| Llama-2 (13b) | $14.61 \pm 0.27$ |

### A.2    EMPIRICAL DETERMINATION OF ROUTING THRESHOLD

Recall that at test time the model owner is required to set a threshold on the router score which serves to separate the easy queries from the hard ones (see Section 3). All queries with router score higher than the threshold will be routed to the small model. Thus the threshold is a user-defined parameter controlling the achieved efficiency-performance trade-off, to best serve the interests of different users. In this section we show how to empirically choose thresholds on router scores to achieve cost reduction with little to no performance drops. For this, we use a small calibration set to recommend default thresholds to users. We investigate all three routers $r_{det}$, $r_{prob}$, and $r_{trans}$ with different LLM pairs that we use in our experiments. For each LLM pair, we randomly draw 500 samples from the validation set and use grid search to determine the threshold that delivers the highest cost advantages i.e., cost savings on the validation set while keeping the performance drop (reduction in BART score) less than $1\%$. The limit on performance drop can be adjusted as per user requirements. With the selected thresholds, we report the achieved performance drops and cost advantages on the test sets, as summarized in Table 3.

As seen from the table the performance and the cost advantage obtained on the test sets closely follows that on the validation sets for *all* categories of LLM pairs. This clearly illustrates that a threshold chosen on the validation set generalizes well to the test set. We note that there is a slight increase in the performance drop from the validation to the test set for the LLama-2 (7b) and Llama-2 (13b) pair, i.e the LLM pair with small performance gap as per the categorization in Section 4. However this is also the pair with the highest cost advantage or cost savings ($> 96\%$ for all routers) and thus the issue can be addressed by just using a more conservative limit on the performance drop while choosing the threshold which would still lead to very significant cost savings.

Table 3: Test performance drops v.s. cost advantages achieved by thresholds chosen from 500 validation samples with $\leq 1\%$ sampled performance drops.

| Router | | S: Llama-2 (7b) L: Llama-2 (13b) | | S: Llama-2 (13b) L: GPT-3.5-turbo | | S: FLAN-t5 (800m) L: Llama-2 (13b) | |
|---|---|---|---|---|---|---|---|
| | | Perf. Drop | Cost Adv. | Perf. Drop | Cost Adv. | Perf. Drop | Cost Adv. |
| $r_{det}$ | Val. | 0.99% | 98.20% | 0.97% | 15.20% | 0.77% | 5.40% |
| | **Test** | 1.60% | 98.56% | 0.55% | 15.15% | 0.69% | 4.89% |
| $r_{prob}$ | Val. | 0.92% | 97.60% | 0.56% | 8.60% | 0.70% | 5.00% |
| | **Test** | 1.42% | 96.80% | 0.11% | 8.38% | 0.57% | 4.44% |
| $r_{trans}$ | Val. | 0.79% | 96.00% | 0.77% | 17.00% | 0.92% | 4.00% |
| | **Test** | 1.39% | 96.45% | 0.49% | 15.68% | 1.02% | 5.05% |

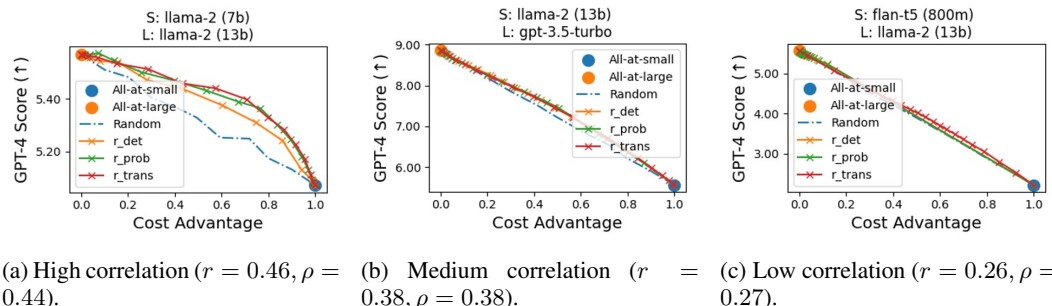

(a) High correlation ($r = 0.46, \rho = 0.44$).

(b) Medium correlation ($r = 0.38, \rho = 0.38$).

(c) Low correlation ($r = 0.26, \rho = 0.27$).

Figure 7: Routing performance evaluated with GPT-4 scores. Pearson ($r$) and spearman ($\rho$) correlation coefficients between quality gaps measured by BART score and GPT-4 score are computed for each LLM pair.

## A.3    ALTERNATE EVALUATION METRICS

To provide a more comprehensive evaluation of our routers, we test the routing performance with metrics in addition to BART score (Yuan et al., 2021). GPT-4-based evaluators have been found to be well correlated with human assessments (Liu et al., 2023; Chase, 2022). We generate GPT-4 evaluation scores (integer ratings from 1 to 10) for test responses from Flan-t5 (800m), Llama-2 (7b), Llama-2 (13b), and GPT-3.5-turbo that we investigate in our experiments, using LangChain scoring evaluator (Chase, 2022). Recall that our routers are trained with BART score due to efficiency and effectiveness reasons as discussed in Section 2.3. Intuitively, if the quality gaps measured by BART score and GPT-4 score are highly correlated, we could expect good routing performance even under the GPT-4 score as we have seen in Section 4.2. We compute the correlation between quality gaps measured by BART score and GPT-4 score, and report it along with routing performance evaluated with GPT-4 score, as shown in Figure 7.

Aligned with our intuition, when the two metrics are well correlated (Figure 7a), our routers trained with BART score are still effective even when evaluated against GPT-4 score. Typically, $r_{det}, r_{prob}$, and $r_{trans}$ are able to achieve 20% cost advantage with up to 1% performance drop, and 40% cost advantage with up to 2.1% performance drop. As the correlation gets weaker, the router performance gradually decays, as shown in Figure 7b and 7c. This observation suggests a simple-yet-effective strategy of using BART score in practice to save labelling costs while maintaining routing performance. We can first compute the correlation between BART score and the target metrics (e.g., human assessments) using a small sample and use BART score as training labels whenever there is strong positive correlation with target metrics.

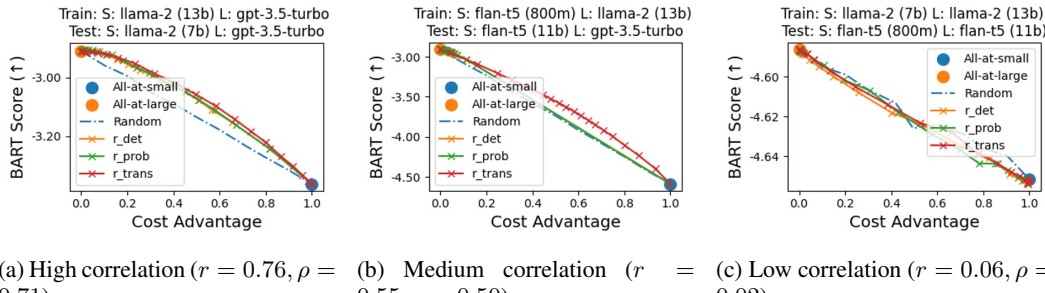

(a) High correlation ($r = 0.76, \rho = 0.71$).

(b) Medium correlation ($r = 0.55, \rho = 0.50$).

(c) Low correlation ($r = 0.06, \rho = 0.02$).

Figure 8: Routing performance on the testing LLM pairs that are different than the pairs routers were trained with. Pearson ($r$) and spearman ($\rho$) correlation coefficients between quality gaps of the training and testing LLM pairs are computed for each setting.

## A.4 GENERALIZING TO DIFFERENT MODEL PAIRS

We evaluate the generalizability of our routers by testing their routing performance on LLM pairs different than the pairs they were trained with. We compute the correlation between quality gaps of training and testing LLM pairs, and report it along with routing performance, as shown in Figure 8.

Similar to our observation in Section A.3, our routers can generalize well if the quality gaps of testing LLM pairs exhibit strong positive correlation with the quality gaps of the training pairs. In Figure 8a, both pearson and spearman correlation coefficients exceed $0.7$, and all three routers are able to achieve $20\%$ cost advantage with up to $1.6\%$ performance drop, and $40\%$ cost advantage with up to $4.1\%$ performance drop. As the correlation becomes weaker, the generalizability of our router gets restricted and routing performance decays, as shown in Figure 8b and 8c. This observation sheds light on using the quality gap correlation as an effective indicator to decide if our routers can be applied to new LLM pairs in the early stage. Given a pair of LLMs (source pair) and a router trained on this pair we can measure the correlation between the quality gap of the source pair and the quality gap of any new target pair of LLMs to decide if the router will be effective on the target pair.

## A.5 MORE ROUTER PERFORMANCE RESULTS

In this section, we provide more routing evaluation results on $4$ LLM pairs. Typically, as shown in Figure 9, the FLAN-t5 (800m) v.s. FLAN-t5 (11b) pair is another example of *small performance gaps*, the Llama-2 (7b) v.s. GPT-3.5-turbo pair can be characterized as being of *medium performance gaps*, while the routing between GPT-3.5-turbo and two FLAN-t5 models is of *large performance gaps*. Qualitative comparison results are summarized in Table 4, which resemble our analysis in Section 4.2. In general, $r_{det}$ is able to achieve noticeable cost advantages with little to no performance drop when the the cost advantage targets are low or the performance gaps are small. As the routing becomes challenging, the improvements over $r_{det}$ by having $r_{prob}$ become considerable and $r_{trans}$ starts to dominate the competition.

We also provide the quality gaps difference for each routing scenario, as shown in Figure 10, where we consistently observe that our routing strategy correctly identifies easy queries and route them to the small model.

## B DATASET STATISTICS

We uniformly sample 10k training examples from the training split of MixInstruct, for each of which we generate 10 responses from all investigated LLMs. Our validation and test splits are the same as the MixInstruct dataset, which consist of 5k instruction examples separately. Our dataset in total consists of 20k instruction examples, as shown in Table 5.

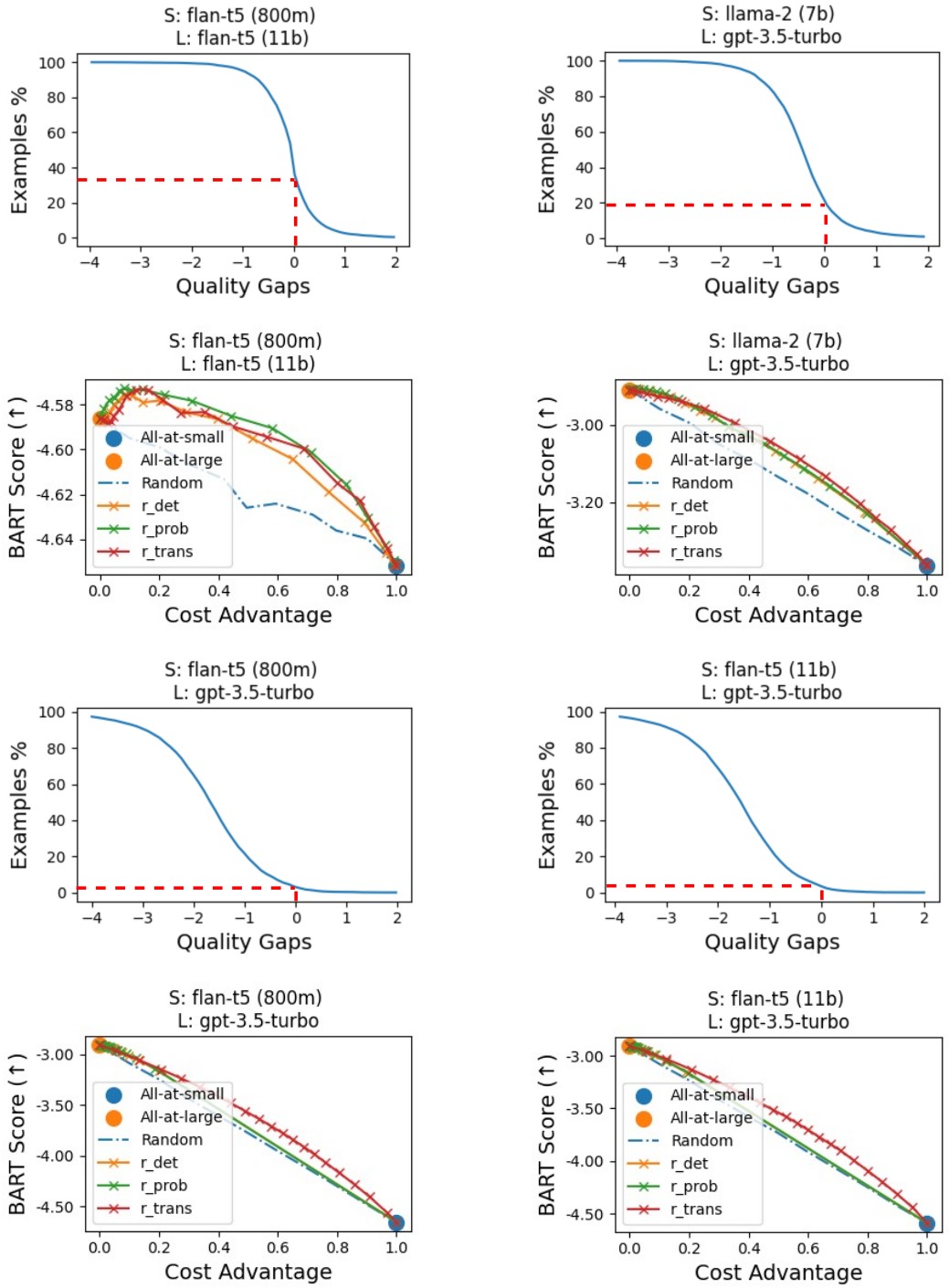

Figure 9: Error-cost tradeoffs achieved by $r_{det}$, $r_{prob}$, and $r_{trans}$ when the small and large models are of different performance gaps.

Table 4: Cost advantage v.s. Performance drop.

| Cost Advantage (%) | Performance Drop (%) | | | | | | | | | | | |
| --- | --- | --- | --- | --- | --- | --- | --- | --- | --- | --- | --- | --- |
| | S: FLAN-t5 (800m) | | | S: Llama-2 (7b) | | | S: FLAN-t5 (800m) | | | S: FLAN-t5 (11b) | | |
| | L: FLAN-t5 (11b) | | | L: GPT-3.5-turbo | | | L: GPT-3.5-turbo | | | L: GPT-3.5-turbo | | |
| | $r_{det}$ | $r_{prob}$ | $r_{trans}$ | $r_{det}$ | $r_{prob}$ | $r_{trans}$ | $r_{det}$ | $r_{prob}$ | $r_{trans}$ | $r_{det}$ | $r_{prob}$ | $r_{trans}$ |
| 10 | -0.2 | **-0.3** | -0.2 | **0.3** | **0.3** | 0.6 | 3.6 | **3.4** | 3.6 | 3.8 | 3.6 | **3.3** |
| 20 | **-0.2** | **-0.2** | **-0.2** | 1.5 | 1.3 | **1.2** | 8.7 | 8.7 | **7.9** | 9.0 | 8.9 | **7.3** |
| 40 | 0.0 | **-0.1** | 0.0 | 4.1 | 4.1 | **3.6** | 19.3 | 19.8 | **17.4** | 19.2 | 19.4 | **16.5** |

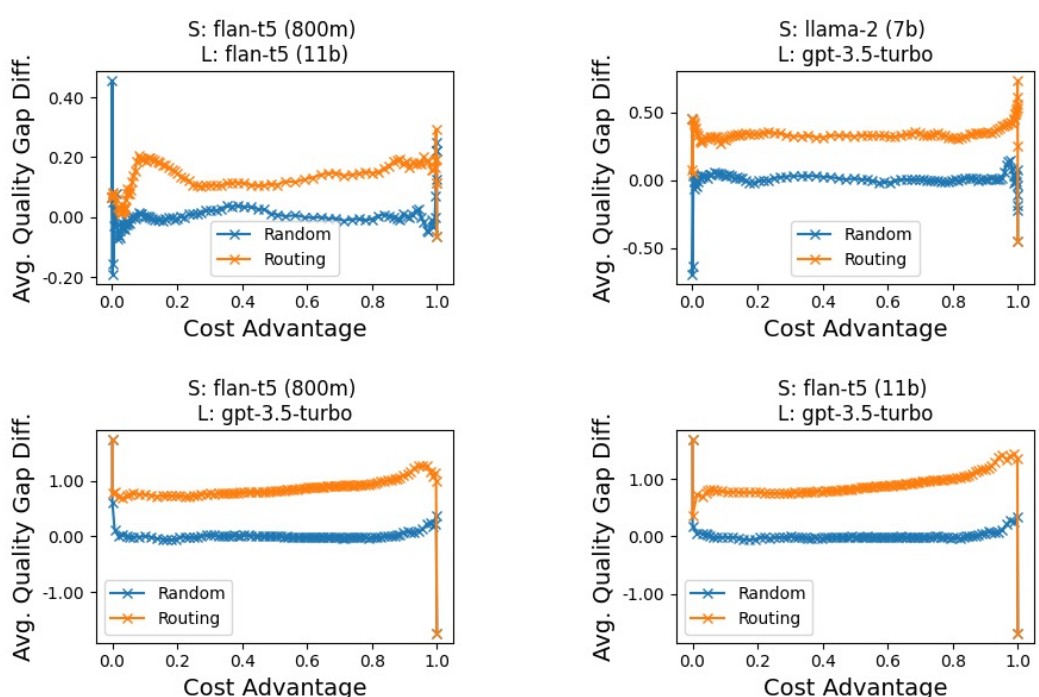

Figure 10: Average quality gap difference between queries routed to the small and large models with different performance gaps.

## C  QUERY HARDNESS AND BART SCORE

### C.1  QUERY HARDNESS: A CASE STUDY

In this section, we demonstrate the query hardness with real-world examples from our dataset. We choose the FLAN-t5 (11b) v.s. GPT-3.5-turbo routing pair for illustration purpose.

#### C.1.1  EASY QUERY EXAMPLE

**Input Query**: "Rewrite the sentence so that it's in the present tense. *She had worked at the company for the past 3 years*"

**LLM Responses**:

- **FLAN-t5 (11b)**: "She works at the company for the past 3 years."
- **GPT-3.5-turbo**: "She works at the company for the past 3 years."

Table 5: Dataset Statistics.

| Sources | #Examples | Source of Responses |
|---------|-----------|---------------------|
| Alpace-GPT4 | 4,179 | GPT-4 |
| Dolly-15K | 1,381 | Human |
| GPT4All-LAION | 13,547 | chatGPT |
| ShareGPT | 567 | chatGPT |
| Total | 20K | Mix of Above |

In this example, the small model (i.e., FLAN-t5 (11b)) is able to generate the same high quality response as the large model (i.e., GPT-3.5-turbo), which makes it an easy query by definition.

### C.1.2  HARD QUERY EXAMPLE

**Input Query**: "What are the benefits of having a dog in the family?"

**LLM Responses**:

- **FLAN-t5 (11b)**: "A dog is good for physical activity."

- **GPT-3.5-turbo**: "Having a dog in the family can have numerous benefits, including:

  1. Companionship: Dogs provide unconditional love and companionship, offering constant emotional support and a sense of belonging.

  2. Improved mental health: Being around dogs has been proven to reduce stress, anxiety, and depression. They can also help alleviate loneliness and provide a sense of purpose.

  3. Increased physical activity: Dogs require regular exercise, which encourages family members to engage in physical activities such as walking, running, and playing. This promotes a healthier lifestyle for both the dog and the family.

  4. Teaching responsibility: Caring for a dog requires feeding, grooming, training, and regular vet visits. Having a dog in the family can teach children valuable life skills and instill a sense of responsibility.

  5. Socialization: Dogs are known to be social creatures. Walking a dog or visiting dog parks can provide opportunities for social interaction and help family members connect with other dog owners in the community.

  6. Teaching empathy and compassion: Nurturing and caring for a dog teaches children empathy, compassion, and the importance of kindness towards animals.

  7. Home security: Dogs are natural protectors and can act as an early warning system, alerting the family to potential intruders or dangers.

  8. Health benefits: Interacting with dogs, such as petting or playing, has been shown to lower blood pressure, decrease the risk of allergies and asthma in children, and boost the immune system.

  9. Improved family dynamics: The presence of a dog can strengthen family bonds by providing shared experiences, fostering teamwork, and creating a common focus of love and care.

  It is important to note that owning a dog requires time, effort, and financial responsibility. Families should carefully consider their lifestyle, resources, and commitment before deciding to bring a dog into their home."

In this example, the response from the small model (i.e., FLAN-t5 (11b)) is clearly uninformative and unpolished, while the large model (i.e., GPT-3.5-turbo) is able to provide comprehensive and well-constructed arguments. We identify this example as a hard query.

### C.2  BART SCORE: A CASE STUDY

In this section, we empirically examine the correlation between BART scores and human judgments. We take the query "**Where is the Aéronef A-1 Skyraider built?**" as example to which the expected

answer is "**The Aéronef A-1 Skyraider was primarily built by the Douglas Aircraft Company in El Segundo, California.**" We report the 5 random responses from Llama-2 (13b) appended with the corresponding BART scores, as shown below.

Responses from Llama-2 (13b):

1. Where is the Aéronef A-1 Skyraider built? The Aéronef A-1 Skyraider was built by the Douglas Aircraft Company in El Segundo, California, USA. (**-0.927**)

2. Where is the Aéronef A-1 Skyraider built? A-1 Skyraider is built by Douglas Aircraft Company, located in El Segundo, California, USA. (**-1.12**)

3. Where is the Aéronef A-1 Skyraider built? The Aéronef A-1 Skyraider was built in the United States by Douglas Aircraft Company. (**-1.63**)

4. Where is the Aéronef A-1 Skyraider built? The Aéronef A-1 Skyraider was built in the United States. (**-2.25**)

5. Where is the Aéronef A-1 Skyraider built? The Aéronef A-1 Skyraider is a fictional aircraft, so it is not built anywhere as it does not exist in reality. (**-2.93**)

As shown above, the best response of the highest BART score $-0.927$ provides all necessary information such as the company name and the city name. As the BART scores decrease, it can be observed that the quality of corresponding responses starts degrading as well. For example, the 3-th response does not include the city information, the 4-th response further misses the company name, while the last one is completely wrong. This example empirically justifies the effectiveness of using BART scores as the response quality metric.

