# OpenReview forum: "Hybrid LLM: Cost-Efficient and Quality-Aware Query Routing"
_ICLR.cc/2024/Conference — ICLR 2024 poster_

### Official Review · Reviewer_ZnPu · 2023-10-29

**Soundness:** 2 fair
**Presentation:** 3 good
**Contribution:** 2 fair
**Rating:** 6
**Confidence:** 2

**Summary:**

Deployment of large language models is costly whereas smaller models can be deployed on edge devices but tend to lag behind in response quality. This work proposes a hybrid inference approach to save cost and maintain response quality. A query router is employed for assigning queries to large or small language model depending upon the predicted query difficulty and the desired quality level. This desired quality level is dynamically tunable at test time for trading quality for cost. The proposed design achieves 40% fewer calls to large model with no drop in response quality.

**Strengths:**

This paper addresses and interesting problem considering that currently available smaller language models can fairly perform well. Depending upon the predicted difficulty level of the query, it is interesting to use a router to pass the relatively easier queries to smaller model. This approach can be cost effective.

Multiple router score designs are proposed.

There is thorough empirical analysis with good discussion.

**Weaknesses:**

The proposed design requires that for each LLM pair, a router is required to be trained which might be a costly undertaking in a production environment.

This paper discusses the cost/quality analysis in context of a language model pair. In real world scenarios, there might be multiple LLMs available and several competing factors to be optimized or traded-off.

Figure 1 is not properly aligned and some inconsistent border is visible (Fig 1 (C)).

**Questions:**

This Paper states that “we expect that using the router to route queries to the small model will not detract significantly from the realizable cost advantage.” Is this an assumption or empirically verified conclusion?

---

> ### Author Response · Authors · 2023-11-17
> **To Reviewer ZnPu**
>
> We thank you for your reviews and address your concerns as follows.
>
> Q1: The proposed design requires that for each LLM pair, a router is required to be trained which might be a costly undertaking in a production environment.
>
> A1: Thank you for this insightful comment. Though our routers are trained for each LLM pair, the learnt knowledge can be generalized to different LLM pairs of similar performance gaps. We will conduct experiments to examine if our trained routers can be applied to different LLM pairs from the one on which it was trained while still being effective. These experiment results will be added to the revised manuscript soon.
>
> Q2: This paper discusses the cost/quality analysis in context of a language model pair. In real world scenarios, there might be multiple LLMs available and several competing factors to be optimized or traded-off.
>
> A2: Thank you for this valuable comment. Routing between multiple LLMs is an important and natural extension to our formulation, which we have briefly discussed in Section 5. To the best of our knowledge, we are the first to explore cost-effective and quality-aware hybrid LLM inference. With our work in this paper on 2-model routing, we believe we have laid the ground for future research on N-model routing.
>
> Q3: Figure 1 is not properly aligned and some inconsistent border is visible (Fig 1 (C)).
>
> A3: Thank you for this comment. We will fix the presentation issues in our revision.
>
> Q4: This Paper states that “we expect that using the router to route queries to the small model will not detract significantly from the realizable cost advantage.” Is this an assumption or empirically verified conclusion?
>
> A4: Thank you for the comment. This claim has been empirically verified. Since our router is an encoder model of a relatively small size (i.e., 300M) in comparison to LLMs we investigated in this work (e.g., Llama-2-13B), the computation overhead of using the router is negligible. We will further empirically validate this claim by comparing the inference latency of our router and different LLMs and add the results of those experiments to the revision.
>
> Thank you for your time and consideration. We sincerely hope that you would consider increasing your rating if you find our responses helpful.

---

> > ### Author Response · Authors · 2023-11-22
> > **To Reviewer ZnPu - Revision is Ready**
> >
> > Thank you very much for your reviews. Addressing your points enabled us to greatly improve the quality of the paper.
> >
> > - In Appendix A.1 (Page 13), we have included inference latency of our router and LLMs investigated in this paper to show that the compute overhead introduced by routing is negligible.
> > - In Appendix A.4 (Page 14), we have included generalizability experiments to show that our routers can still be effective with LLM pairs different from the pairs they were trained with. These experiments show that as long as the quality gap (defined in Section 3) of the two pairs are correlated (which can be checked on a validation set), a router trained on one pair can be effective even on the other pair. This clearly illustrates how the router can be generalized across LLM pairs.
> > - We have fixed the presentation issues in Figure 1 (Page 2).
> >
> > We hope we have answered your questions to your satisfaction and hope you would consider increasing the score.

---

### Official Review · Reviewer_VA7E · 2023-10-29

**Soundness:** 3 good
**Presentation:** 3 good
**Contribution:** 2 fair
**Rating:** 6
**Confidence:** 2

**Summary:**

The paper introduces a novel inference paradigm called hybrid inference, which utilizes two models of different sizes to handle queries. This approach aims to balance infference cost and response quality by routing easy queries to a smaller model while directing more complex queries to a larger model. The authors propose an orchestration framework that involves a router trained on a dataset of representative queries. The router dynamically routes queries to the appropriate model, thus reducing overall costs while maintaining response quality. They present three variations of the router: a deterministic router, a probabilistic router, and a probabilistic router with data transformation.

**Strengths:**

The paper sets the problem in the context of LLM inference and focuses on the evaluation of response quality and cost advantage. It defines metrics for measuring the effectiveness of the routing strategy, considering the intrinsic uncertainties in natural language processing tasks. The evaluation is conducted on the MixInstruct dataset, which comprises a diverse range of tasks such as question answering, summarization, and information extraction. The experimntal results demonstrate the efficacy of the proposed routing strategies, especially in scenarios where the performance gap between the small and large models is minimal. The deterministic router achieves good cost advantages with negligible drops in response quality, while the probabilistic router further improves the cost advantage without compromising response quality. The probabilistic router with data transformation exhibits even more promising results, achieving significant cost advantages with no quality drop.

**Weaknesses:**

The main limitation of the paper seems to be its reliance on the assumptions about the quality gaps and the routiing mechanisms. These assumptions could potentially affect the overall effectiveness and efficiency of the routing process. Additionally, the reliance on specific models and the need for manual intervention in setting the threshold for routing may limit the scalability and generalizability of the proposed framework.

**Questions:**

The approach might encounter problems in accurately distinguishing between easy and hard queries, especially when dealing with a large performance gap between different models. How do you elaborate on this?

---

> ### Author Response · Authors · 2023-11-17
> **To Reviewer VA7E**
>
> We thank you for your reviews and address your concerns as follows.
>
> Q1: The main limitation of the paper seems to be its reliance on the assumptions about the quality gaps and the routing mechanisms. These assumptions could potentially affect the overall effectiveness and efficiency of the routing process.
>
> A1: Thank you for this comment. First of all, we would like to ask for clarification on what assumptions you are referring to, given that we did not have any formal assumptions “about the quality gaps and the routing mechanisms” in our paper.
>
> Quality gaps between responses from different LLMs is not an assumption but an important and real observation that has been widely reported and studied in previous work [1,2]. In general, LLMs of larger model sizes, trained with more data and computational resources have been found to give responses of higher quality [1,2]. The setting in which responses from LLMs are of negligible quality difference is an extreme case, and a degenerate solution which always routes queries to the small models will be optimal in this setting. In this work, we study the more general problem where response quality gaps can be significant. Our router is designed to identify easy queries, to which even small models can give high quality responses, and route them to small models to save costs while maintaining high performance.
>
> As to the routing mechanism, we developed 3 routing strategies, r_det, r_prob, and r_trans. The deterministic router, r_det, is the only one relying on the assumption that “neural models are deterministic functions”. In our analysis, we show that this assumption may not hold in practice, which motivates our design of r_prob and r_trans. Both r_prob and r_trans are generic methods for LLM routing and we demonstrated their effectiveness with different LLMs and real-world queries.
>
> Q2: Additionally, the reliance on specific models and the need for manual intervention in setting the threshold for routing may limit the scalability and generalizability of the proposed framework.
>
> A2: Thank you for this comment. First of all, we would like to clarify that our routing framework is generic and can be applied on any given LLM pairs. In addition, we will conduct experiments to show that our trained routers can be applied to different LLM pairs while still being effective. The results of these experiments will be added into the revised manuscript soon.
>
> Secondly, the validity of our framework does not rely on the choices of thresholds. Instead, the threshold is a user-defined parameter used to control the efficiency-performance trade-off, to best serve the interests of different users. In practice, if needed, we could use a calibration set to recommend by-default thresholds for users to decide. We will illustrate this through extra experiments and report results in the revision.
>
> Q3: The approach might encounter problems in accurately distinguishing between easy and hard queries, especially when dealing with a large performance gap between different models. How do you elaborate on this?
>
> A3: Thank you for this comment. In Section 4.3, we demonstrate that our router can distinguish and route easy queries to small models while routing hard queries to large models, with LLMs of small, medium, and large performance gaps. In Figure 6, we plot the difference between the average quality gaps of queries routed to the small model and those routed to the large model for our router and the random baseline w.r.t. different values of cost advantages (i.e., the fraction of queries routed to the small model). The random baseline randomly assigns queries and therefore the average difference is nearly always zero. In contrast, the difference between the average quality gaps for our router always has significant positive values at all cost advantages for all LLM pairs, which indicates that our approach can successfully distinguish easy queries (i.e., queries of large quality gaps (i.e., q(S(x)) - q(L(x)))), and route more easy queries to the small model as expected.
>
> Moreover, though the routing performance may vary on different LLM pairs due to the intrinsic difference of routing settings, our approach is shown to consistently outperform competing methods with non-trivial performance gains, as discussed in Section 4.2.
>
> Thank you for your time and consideration. We sincerely hope that you would consider increasing your rating if you find our responses helpful.
>
> References:
> [1] Hoffmann, Jordan, et al. "Training compute-optimal large language models." arXiv preprint arXiv:2203.15556 (2022).
> [2] Kaplan, Jared, et al. "Scaling laws for neural language models." arXiv preprint arXiv:2001.08361 (2020).

---

> > ### Author Response · Authors · 2023-11-22
> > **To Reviewer VA7E - Revision is Ready**
> >
> > We thank you again for your suggestions which were extremely helpful in making the router more useful in practice. We have added the following experiments to address your points:
> >
> > - In Appendix A.2 (Page 13), we have demonstrated how to empirically choose the router thresholds using a small validation set. We choose the threshold by checking the performance and the cost savings with different thresholds on a small validation set and choosing the one that satisfies the user requirements. We then show that the performance and the cost savings are maintained even when using this threshold on the test set thus illustrating an effective way of determining the threshold in practice.
> > - In Appendix A.4 (Page 14), we have included generalizability experiments to show that our routers can still be effective with LLM pairs different from the pairs they were trained with. These experiments show that as long as the quality gap (defined in Section 3) of the two pairs are correlated (which can be checked on a validation set), a router trained on one pair can be effective even on the other pair. This clearly illustrates how the router can be generalized across LLM pairs.
> >
> > We hope we have answered your questions to your satisfaction and hope you would consider increasing the score.

---

### Official Review · Reviewer_uyT6 · 2023-11-08

**Soundness:** 2 fair
**Presentation:** 2 fair
**Contribution:** 2 fair
**Rating:** 3
**Confidence:** 4

**Summary:**

The paper introduces a hybrid inference strategy designed to minimize the computational expense by limiting the number of queries to the larger model and utilizing smaller models to function as decision-making routers.
Initially, the approach assesses if the user's input query is easy or hard by evaluating the anticipated response quality from both the small and large models.
To evaluate the complexity of a query, the paper describes three distinct methodologies, each utilizing the same classification model but differing in their training and inference schemes.

**Strengths:**

Paper presents a novel hybrid inference strategy designed to minimize the computational expense by limiting the number of queries to the larger model and utilizing smaller models to function as decision-making routers.
Moreover, paper presents multiple different approaches to training the decision making routers and its effectiveness.

**Weaknesses:**

I have following major concerns.

1. **Reliability of BART scores for routing**
I am uncertain about the efficiency of training the router model to decide whether the BART scores of the smaller model is similar to those of the larger one. BARTScore has demonstrated strong performance in extractive QA; however, its correlation may diminish in abstractive QA contexts [1], suggesting that the metric might not be suitable for assessing open-ended generation tasks. Establishing a correlation between evaluations of routing using BARTScore and human assessments would be beneficial to verify the reliability of BARTScore for routing evaluation purposes.

2. **The Impact of Training Data Versus Model Size**
I am of the opinion that the size of the model is not as critical as the differences in the training data used for each model in determining quality. For instance, consider evaluating the performance disparities between models like (Llama-2 7B and Llama-2 13B) versus those between (Llama-2-7B and the more recent Zephyr-7B [2]). Would the performance gap trend similar to the reported trend in Figure6?

[1] G-EVAL: NLG Evaluation using GPT-4 with Better Human Alignment., Liu et al., 2023 \
[2] https://huggingface.co/HuggingFaceH4/zephyr-7b-alpha

**Questions:**

Same as weakness part.

---

> ### Author Response · Authors · 2023-11-17
> **To Reviewer uyT6 (Part I)**
>
> We thank you for your reviews and address your concerns below.
>
> Q1: Reliability of BART scores for routing. I am uncertain about the efficiency of training the router model to decide whether the BART scores of the smaller model is similar to those of the larger one. BARTScore has demonstrated strong performance in extractive QA; however, its correlation may diminish in abstractive QA contexts [1], suggesting that the metric might not be suitable for assessing open-ended generation tasks. Establishing a correlation between evaluations of routing using BARTScore and human assessments would be beneficial to verify the reliability of BARTScore for routing evaluation purposes.
>
> A1: Thank you for this insightful comment. We address this concern from the following perspectives:
> 1. Firstly, we would like to mention that the design of our routing strategies is agnostic to the choice of response quality metrics. In principle, our router can be trained with any text-generation metrics.
> 2. Secondly, we demonstrate our approaches using BARTScore for two reasons. (1) BARTScore is inexpensive to compute in comparison to human and GPT-based evaluators [1,11] due to its small model size (e.g., BART-base is of 140M parameters), which makes it a more accessible choice especially with large training data. (2) BARTScore has been found to have good correlation with GPT-Rank [2], a strong GPT-based ranking metric, on MixInstruct dataset which includes open-ended question answering and creative writing tasks. Specifically, in [2], the authors found that “BARTScore gets the highest correlation with GPT-Rank against other metrics, which suggests we use BARTScore to provide supervision for training”. We follow this recent work by choosing BARTScore for router training and evaluation purposes.
> 3. Thirdly, we would like to bring up that [1] and [2] are using BARTScore in two different ways which may explain their different observations on the effectiveness of BARTScore, especially on open-ended generation tasks. The general idea behind BARTScore is that “models trained to convert the generated text to/from a reference output or the source text will achieve higher scores when the generated text is better” [10]. With this in mind we make the following observations:
>     1. In [1], BARTScore measures the quality of the machine-generated summary in terms of the likelihood of obtaining the summary from the source text. The authors compare the quality as measured by the BARTScore and the quality as measured by GPT-4 (which is directly asked to rate the summary) and find that the latter is more correlated with the quality as measured by human evaluators.
>     2. In [2], BARTScore is used to compare the responses from different LLMs to the same query in terms of the likelihood of obtaining the given LLM response from a ground truth (human/GPT-4 generated response). As a baseline the authors also use ChatGPT to compare the two responses. In this case, the authors find that the BARTScore measurement is well correlated with the ChatGPT measurement i.e. the two typically provide the similar ranking of the LLM responses.
>
>     From the above observations we believe it can be concluded that the observations from [1] and [2] are not against but rather complementary to each other. A holistic view would be that, **the values of BARTScore may not correlate well with human assessments on open-ended generation tasks (as seen in [1]), but it is informative in ranking multiple responses to the same query (as seen in [2]) while being significantly cheaper to compute than human evaluation or GPT ranking**. Since the goal of this work is to route queries to the smaller LLM when we expect its response to be comparable to or better than that of the larger LLM, therefore we believe that our setting is closer to [2] and using a metric like BARTScore that is well correlated with the ranking of LLM responses is sufficient for our purpose. Overall, we do not claim that BARTScore is the best metric for all text-generation tasks, but argue that it is an empirically good fit for our routing task, due to the reasons discussed above.
> 4. Last but not least, we will evaluate our approach with more effective metrics in addition to BARTScore. Since human assessment of semantic interpretation tasks by itself is a challenging problem, due to the subjectivity and inconsistency of human annotators among themselves [3,4,5,6], we are going to use GPT-based evaluators and report evaluation results in our revision.

---

> > ### Author Response · Authors · 2023-11-17
> > **To Reviewer uyT6 (Part II)**
> >
> > Q2: The Impact of Training Data Versus Model Size. I am of the opinion that the size of the model is not as critical as the differences in the training data used for each model in determining quality. For instance, consider evaluating the performance disparities between models like (Llama-2 7B and Llama-2 13B) versus those between (Llama-2-7B and the more recent Zephyr-7B [7]). Would the performance gap trend similar to the reported trend in Figure6?
> >
> > A2: Thank you for this valuable comment. We address this concern as follows.
> > 1. Both model sizes and training data are important to model quality as shown in recent scaling law studies [8,9]. Notably, our routing strategies are agnostic to the sources of model performance disparities. In our paper, we reported results for LLM pairs of different sizes and/or trained on different data (e.g., Llama-2-13B v.s. GPT-3.5-turbo and Flan-t5-800m v.s. Llama-2-13B).
> > 2. More importantly, in this work, we are interested in reducing inference costs while maintaining high performance. For this purpose, we mainly consider models of significant cost differences. The setting that models are of similar sizes but trained with different data (e.g., Llama-2-7B v.s. Zephyr-7B) is an extreme case where we can hardly expect any cost savings, given that the two models are of similar sizes as well as costs.
> > 3. The similar-size-but-different-data setting is essentially sensible when the goal is to improve overall performance while maintaining almost constant inference latency such as the load-balancing scenarios. This is an interesting problem but with a different goal than our formulation.
> >
> > Thank you for your time and consideration. We sincerely hope that you would consider increasing your rating if you find our responses helpful.
> >
> > References:
> > [1] G-EVAL: NLG Evaluation using GPT-4 with Better Human Alignment., Liu et al., 2023.
> > [2] Jiang, Dongfu, Xiang Ren, and Bill Yuchen Lin. "LLM-Blender: Ensembling Large Language Models with Pairwise Ranking and Generative Fusion." arXiv preprint arXiv:2306.02561 (2023).
> > [3] Davani, Aida Mostafazadeh, Mark Díaz, and Vinodkumar Prabhakaran. "Dealing with disagreements: Looking beyond the majority vote in subjective annotations." Transactions of the Association for Computational Linguistics 10 (2022): 92-110.
> > [4] Denton, Emily, et al. "Whose ground truth? accounting for individual and collective identities underlying dataset annotation." arXiv preprint arXiv:2112.04554 (2021).
> > [5] Aroyo, Lora, and Chris Welty. "Truth is a lie: Crowd truth and the seven myths of human annotation." AI Magazine 36.1 (2015): 15-24.
> > [6] Aroyo, Lora, et al. "DICES Dataset: Diversity in Conversational AI Evaluation for Safety." arXiv preprint arXiv:2306.11247 (2023).
> > [7] https://huggingface.co/HuggingFaceH4/zephyr-7b-alpha.
> > [8] Hoffmann, Jordan, et al. "Training compute-optimal large language models." arXiv preprint arXiv:2203.15556 (2022).
> > [9] Kaplan, Jared, et al. "Scaling laws for neural language models." arXiv preprint arXiv:2001.08361 (2020).
> > [10] Yuan, Weizhe, Graham Neubig, and Pengfei Liu. "Bartscore: Evaluating generated text as text generation." Advances in Neural Information Processing Systems 34 (2021): 27263-27277.
> > [11] Fu, Jinlan, et al. "Gptscore: Evaluate as you desire." arXiv preprint arXiv:2302.04166 (2023).

---

> > > ### Author Response · Authors · 2023-11-22
> > > **To Reviewer uyT6 - Revision is Ready**
> > >
> > > We thank you again for suggesting the use of GPT-4 as an alternative evaluator for our approach. This was a great suggestion as it enabled us to explore the situation where the router may be evaluated on a different metric from the one which it is trained on which may often be the case in practice. We have added results for this case in Appendix A.3 (Page 13) where we show that there indeed exist scenarios where the BART score is well correlated with the GPT-4 score and thus in such scenarios a router trained on the BART score performs well even under GPT-4 evaluation. While the two scores are not always well correlated, we have outlined the practical advantages of the BART score (combination of low cost and good agreement with human judgement in many scenarios) which make it a better choice for training the router since evaluating large training datasets with GPT-4 or human evaluators may be prohibitively expensive for most researchers/developers. The new results will allow the router owners to highlight the cases where BART and GPT scores are well correlated (by checking on a calibration set) and this information can allow users to decide when and how to use and evaluate the router.
> > >
> > > We hope we have answered your questions to your satisfaction and hope you would consider increasing the score.

---

### Official Review · Reviewer_9pHQ · 2023-11-11

**Soundness:** 4 excellent
**Presentation:** 4 excellent
**Contribution:** 3 good
**Rating:** 8
**Confidence:** 5

**Summary:**

The authors introduce a router that assigns queries to differently sized models. Their method results in 40% fewer calls to the large model, with no drop in response quality. They introduce two main techniques to improve performance:

- using soft probabilities instead of hard probabilities
- using a data transformation with a relaxation $t$ to provide stronger training signal.

**Strengths:**

- Paper is well written, and authors do a good job of building upon concepts used in final technique.
- Ablations and analysis are extensive and well-thought out, giving researchers ample inspiration to build upon this technique.
- The analysis of performance on different model size pairs is interesting to me.

**Weaknesses:**

Please cite these works:
- https://arxiv.org/abs/2305.05176 - routing on a query level
- https://arxiv.org/abs/2211.17192, https://arxiv.org/abs/2302.07863 - latency reduction using small and big models

I believe writing a discussion of the tradeoffs of these approaches would improve the current draft.

**Questions:**

- In this method, we are able to reduce cost and latency, but not as much latency reduction as methods such as speculative decoding (https://arxiv.org/abs/2211.17192). While there is added cost with speculative decoding, do you think there's any possibility of closing this gap?
- Do you think this might be because of scoring query wise vs token-wise? Why not use this method token wise?

---

> ### Author Response · Authors · 2023-11-16
> **To Reviewer 9pHQ**
>
> We thank you for your reviews and address your concerns below.
>
> Q1: Please cite these works: https://arxiv.org/abs/2305.05176 - routing on a query level; https://arxiv.org/abs/2211.17192, https://arxiv.org/abs/2302.07863 - latency reduction using small and big models. I believe writing a discussion of the tradeoffs of these approaches would improve the current draft.
>
> A1: Thank you for pointing out these important references. We will add a discussion of the tradeoffs of these approaches in our revision. In brief,
> 1. FrugalGPT [1] studies how to reduce inference costs using LLM cascades and early-exit strategies. Specifically, for each user query, FrugalGPT will execute a sequence of LLMs and return answers whenever the confidence scores exceed the predefined thresholds. Though empirically effective, FrugalGPT is at the risk of executing multiple LLMs or even all LLMs in the cascade to answer a single query, which could be prohibitively expensive. Instead, our routing mechanism ensures that only one LLM will be used to generate answers for user queries and therefore is likely to be more efficient especially on challenging queries.
> 2. Speculative decoding [2,3] speeds up decoding of expensive models by invoking small-and-efficient decoders on the “easy” decoding steps. Instead, in our work we are interested in query routing which assigns “easy” queries to small models to reduce overall inference costs while maintaining high performance. Clearly, speculative decoding [2,3] optimizes a different objective and is complementary to our approach. A straightforward framework combining the two will be as follows. Users can first use our router to decide if a query should be handled by the small or large model. If the query goes to the small model, we can save significant costs as demonstrated in our paper. If the query goes to the large model, we can still apply speculative decoding to achieve further efficiency improvements.
>
> Q2: In this method, we are able to reduce cost and latency, but not as much latency reduction as methods such as speculative decoding (https://arxiv.org/abs/2211.17192). While there is added cost with speculative decoding, do you think there's any possibility of closing this gap?
>
> A2: Thank you for this question. As discussed in A1, speculative decoding [2, 3] optimizes a different objective than ours, which may explain the difference in achieved cost reduction. Given that the two approaches are complementary, instead of asking how to “close this gap”, it could be more interesting to ask how to achieve further efficiency improvements by combining our approach with speculative decoding. A simple strategy is discussed in A1 and we will investigate other possibilities in the future work.
>
> Q3: Do you think this might be because of scoring query wise vs token-wise? Why not use this method token wise?
>
> A3: Thank you for this question. As discussed above, the difference may largely come from the fact that we are optimizing different objectives than speculative decoding [2,3], with the distinction between query-wise and token-wise objective formulation being one of the contributing factors.
>
> In this work, we did not consider applying our approach on token level mainly for two reasons: (1) We are interested in developing a generic approach that is agnostic to model implementation and generalizes to both open-sourced as well as black-box LLMs. Token-wise routing requires transferring KV values of previous tokens across models which can be infeasible if these intermediate results are unavailable (e.g., black-box LLM APIs) or the dimensions do not match across open-sourced LLMs. (2) A naive token-wise routing requires executing the router at each decoding step, which may lead to notable compute overheads and weaken the achieved cost reduction.
>
> Thank you for your time and consideration. We sincerely hope that you would consider increasing your rating if you find our responses helpful.
>
> References:
> [1] Chen, Lingjiao, Matei Zaharia, and James Zou. "FrugalGPT: How to Use Large Language Models While Reducing Cost and Improving Performance." arXiv preprint arXiv:2305.05176 (2023).
> [2] Leviathan, Yaniv, Matan Kalman, and Yossi Matias. "Fast inference from transformers via speculative decoding." International Conference on Machine Learning. PMLR, 2023.
> [3] Kim, Sehoon, et al. "Speculative Decoding with Big Little Decoder." Thirty-seventh Conference on Neural Information Processing Systems. 2023.

---

> ### Author Response · Authors · 2023-11-22
> **To Reviewer 9pHQ - Revision is Ready**
>
> We thank you again for your reviews and the positive feedback. We greatly appreciate your support of our ideas and have revised the paper as per your suggestions. Specifically, in Section 2.1 (Page 4), we have added a detailed discussion to compare our approach with respect to FrugalGPT [1] and speculative decoding [2,3].
>
> We hope we have answered your questions to your satisfaction and hope you would consider increasing the score.
>
> Reference:
> [1] Chen, Lingjiao, Matei Zaharia, and James Zou. "FrugalGPT: How to Use Large Language Models While Reducing Cost and Improving Performance." arXiv preprint arXiv:2305.05176 (2023).
> [2] Leviathan, Yaniv, Matan Kalman, and Yossi Matias. "Fast inference from transformers via speculative decoding." International Conference on Machine Learning. PMLR, 2023.
> [3] Kim, Sehoon, et al. "Speculative Decoding with Big Little Decoder." Thirty-seventh Conference on Neural Information Processing Systems. 2023.

---

### Author Response · Authors · 2023-11-22
**General Response**

We sincerely thank all reviewers for their valuable comments. We have added a range of new experiments which address the concerns raised by the reviewers and we believe this has made the evaluation of our approach much more comprehensive. The changes are highlighted in blue in the updated draft and are summarized below:

1. We have discussed the trade-offs of related work pointed out by Reviewer 9pHQ in Section 2.1 (Page 4).
2. We have shown that the computational overhead of our router is negligible in Appendix A.1 (Page 13) as suggested by Reviewer ZnPu.
3. We have discussed how to effectively choose the routing threshold in practice using a validation set in Appendix A.2 (Page 13) as suggested by Reviewer VA7E.
4. We have included additional experiments to demonstrate the effectiveness of our router when evaluated under GPT-4 scores instead of the BART score as suggested by Reviewer uyT6. These experiments are presented in Appendix A.3 (Page 13) and show that there are scenarios where the BART score is well correlated with the GPT-4 score and in such scenarios our router is effective even when evaluated as per the GPT-4 score.
5. We have added experiments to demonstrate the generalizability of a router trained on one pair of LLMs to evaluation on a different pair of LLMs in Appendix A.4 (Page 14) as suggested by Reviewer VA7E and Reviewer ZnPu. These experiments show that if the quality gap (defined in Section 3) of the two LLM pairs are well correlated then a router trained on one pair is also effective on the other pair.

The details of revision are referred to the following official comments.

---

### Meta-Review · Area_Chair_G6jL · 2023-12-23

**Metareview:**

This paper proposes a hybrid inference approach that aims to combine the respective strengths of large language models and smaller ones to save cost and maintain quality. The proposed approach adopts a router to assign queries to a small or large model based on the predicted query difficulty and the desired quality level. The reviewers generally agree that the paper is well-written, the exploration of multiple designs and experiments are extensive, and the results are promising. They also point out some weaknesses and limitations, including missing discussions on some related work, reliability of using BART scores for routing, and the generalizability of the proposed approach. During the discussion period, it seems the authors have addressed all comments properly with newly added experiments in appendix. I would recommend the paper to be accepted based on the reviews and discussions.

**Justification For Why Not Higher Score:**

Please see the limitations summarized above. Three out of four reviewers gave positive scores (>=6) but seem not very excited about contributions.

**Justification For Why Not Lower Score:**

Please see the strengths summarized above. In addition, the reviewers have addressed the limitations and comments raised by the reviewers and updated the draft.

---

### Decision · Program_Chairs · 2024-01-16

Accept (poster)